# NH_2_-Functionalized Magnetic Nanoparticles for the *N*-Glycomic Analysis of Patients with Multiple Sclerosis

**DOI:** 10.3390/ijms23169095

**Published:** 2022-08-13

**Authors:** Dalma Dojcsák, Ágnes Mária Ilosvai, László Vanyorek, Ibolya Gilányi, Csaba Oláh, László Horváth, Csaba Váradi

**Affiliations:** 1Advanced Materials and Intelligent Technologies Higher Education and Industrial Cooperation Centre, University of Miskolc, 3515 Miskolc, Hungary; 2Institute of Chemistry, Faculty of Materials Science and Engineering, University of Miskolc, 3515 Miskolc, Hungary; 3Borsod-Abaúj-Zemplén County Center Hospital and University Teaching Hospital, Miskolc, 3526 Miskolc, Hungary

**Keywords:** multiple sclerosis, magnetic nanoparticles, liquid chromatography, mass spectrometry

## Abstract

Glycosylation is vital for well-functioning glycoproteins and is reportedly altered in chronic inflammatory disorders, including multiple sclerosis (MS). High-throughput quantitative measurement of protein glycosylation is challenging, as glycans lack fluorophore groups and require fluorescent labeling. The attachment of fluorescent tags to each glycan moiety necessitates sample clean-up for reliable quantitation. The use of magnetic particles in glycan sample preparation is reportedly an easy-to-use solution to accomplish large-scale biomarker discovery studies. In this study, NH_2_-funtionalized magnetic nanoparticles were synthetized, characterized and applied for the glycosylation analysis of serum samples from patients diagnosed with multiple sclerosis and corresponding healthy controls. Serum samples were PNGase F digested and labeled by procainamide via reductive amination, followed by magnetic nanoparticle-based purification. The prepared samples were analyzed by hydrophilic interaction liquid chromatography, allowing for the relative quantitation of the individual glycan species. Significant glycosylation alterations were detected between MS patients and healthy controls, especially when analyzing the different gender groups.

## 1. Introduction

Multiple sclerosis (MS) is a neurodegenerative autoimmune disorder affecting 3.1 million people worldwide [1]. MS is three times more frequent in women than men, and 50% of people diagnosed with MS are under the age of 30 [2]. The specific leading cause of MS is unknown, although certain factors have been identified as triggers in the development of this disease, such as Epstein–Barr infection, low vitamin D levels, immunological dysfunctions and genetic mutations. Based on current knowledge, the main problem in MS is the abnormal control of the immune system; that is, T cells attack the myelin sheaths of axons in the central nervous system, resulting in inflammation and loss of neurons [3]. The demyelination and subsequent degeneration of axons can result in the physical disability of MS patients, including vision problems, muscle weakness, loss of balance or numbness; thus, rapid and reliable disease identification is critical [4]. In current clinical practice, the differential diagnosis of MS is based on the presented signs and symptoms in combination with medical imaging and laboratory testing, although MS symptoms often overlap with other neurological disorders such as neuromyelitis and neurosarcoidosis [5]. The main signs of MS include the presence of lesions/plaques in the brain and spine, brain atrophy, central vein signs and oligoclonal immunoglobulins in the cerebrospinal fluid [6]. The lack of clear understanding of the mechanism of MS is indicative of the problematic identification of this disorder, as there is no available unique biomarker to provide definitive diagnosis [7].

Protein glycosylation is a critical post-translational modification of serum glycoproteins, reportedly altered in cancer [8] and several chronic inflammatory diseases [9], including MS [10]. The presence of carbohydrate moieties on parent proteins is regulated by numerous glycosyl-transferases and glycosidases; thus, bioanalysis of protein glycosylation can reflect the changes in biochemical processes [11,12]. Altered total plasma glycosylation was reported in MS patients compared to healthy controls, providing substantial value in disease status classification [13]. Aberrant glycosylation of immunoglobulin G (IgG) has also been described in MS patients, suggesting that IgG glycosylation has a significant role in the pathogenesis of MS [14]. While the analysis of protein glycosylation is critical in biomarker discovery research and biopharmaceutical production, its analysis is still challenging due to its complexity and the required sample preparation process [15].

To quantify the glycan species of a given sample, sugars first have to be released from the parent proteins, which is traditionally performed by an enzymatic peptidyl-*N*-glycosidase F (PNGase F) digestion [16]. Due to the lack of fluorophore group, this digestion is followed by the stoichiometric attachment of an aromatic amine (2-aminobenzamide, 2-anthranilic acid, procainamide), most commonly via reductive amination, resulting fluorescently labeled glycans [17]. In order to label each glycan structure in the sample, fluorescent dyes are usually applied in great excess, necessitating a purification step prior to analysis [18]. Strategies to remove excess dye after fluorescent labeling of glycans include solid-phase extraction [19], normal-phase purification [20] and hydrophilic interaction liquid chromatography [21], with the main requirement of sample preconcentration prior to analysis. Magnetic particles are used in several molecular biology applications and bioanalytical sample preparations, with the main advantage of low elution volumes, allowing the bypass of vacuum centrifugation [22]. 

In this study, NH_2_-functionalized magnetic nanoparticles were synthetized, characterized and applied for the quantitative *N*-glycomic analysis of serum samples from patients with MS and healthy controls using hydrophilic interaction liquid chromatography.

## 2. Results and Discussion

### 2.1. Characterization of the Amine-Functionalized Ferrite Nanoparticles

In this study, we synthetized four different magnetic nanoparticles for the selective enrichment of fluorescently derivatized glycans from human serum. Amine functionalization is a traditional method for the isolation of carbohydrates; thus, CoFe_2_O_4_-NH_2_, NiFe_2_O_4_-NH_2_, MgFe_2_O_4_-NH_2_ and MnFe_2_O_4_-NH_2_ were prepared by a coprecipitation method [23]. This was followed by the characterization of the synthetized magnetic nanoparticles and the direct comparison of their performance in glycan purification. The identification of the most suitable magnetic nanoparticle was followed by application on clinical samples, as visualized in Figure 1.

The identification of surface functional groups in the synthetized magnetic nanoparticles was performed by Fourier-transform infrared spectroscopy (FTIR) (Figure 2A). In the infrared spectra, one convoluted band with a shoulder was identified originating from the tetrahedral (at 576 cm^−1^) and octahedral complexes (at 433 cm^−1^) of the metal–oxygen bond vibrations in the spinel structures. The bands at 581 cm^−1^ (NiFe_2_O_4_-NH_2_), 586 cm^−1^ (MnFe_2_O_4_-NH_2_), 567 cm^−1^ (MgFe_2_O_4_-NH_2_) and 595 cm^−1^ (CoFe_2_O_4_-NH_2_) were assigned to the vibrations of Fe^3+^–O^2−^. On the other hand, the bands at the lower wavenumbers (at 417 cm^−1^, 423 cm^−1^, 437 cm^−1^ and 414 cm^−1^) represented the trivalent metal–oxygen vibration at the octahedral B-sites in the NiFe_2_O_4_-NH_2_, MnFe_2_O_4_-NH_2,_ MgFe_2_O_4_-NH_2_ and CoFe_2_O_4_-NH_2_ samples. At the start of the fingerprint region (400–1600 cm^−1^), several vibration bands were located (i.e., νC-N, νC-O, ω (wagging) N-H, δCH_2_). The broad band between 800 cm^−^^1^ and 900 cm^−1^ arose from the νC-O stretching vibration of the alcoholic groups of the ethylene glycol (at 879 cm^−1^), which was adsorbed on the surface of the nanoparticles. The presence of the bands at the 1050 cm^−1^ and 1630 cm^−1^ wavenumbers originated from the stretching vibration of the amine functional groups and the adsorbed ethanolamine molecules on the ferrite surfaces. The low intensive bands between 1370 cm^−1^ and 1410 cm^−1^ were identified as hydroxyl group bending vibrations (βOH). The two miniscule peaks at 2850 cm^−1^ and 2930 cm^−1^ were the symmetric and asymmetric stretching vibration modes of the aliphatic and aromatic C–H bonds. These bands may be the result of the ethylene glycol and ethanolamine derivatives on the particle surface. Finally, the wide absorption band in the region of 3000–3750 cm^−1^ was assigned to the stretching vibration of hydroxyl and amine groups (-NH_2_ and -OH). The hydroxyl and amine functional groups on the surface of the magnetic nanoparticles contribute their dispersibility in polar solvents (Appendix A). Moreover, the aforementioned functional groups tend to form a hydrogen bond with similar groups of polysaccharides, including glycans. Thus, the amine-functionalized ferrite nanoparticles are suitable for the specific isolation of glycans from complex matrices.

During the X-ray powder diffraction (XRD) analysis of the four amine-functionalized ferrites, seven reflection peaks were identified at 18.4° (111), 30.1° (220), 35.4° (311), 43.1° (400), 53.3° (422), 56.8° (511) and 62.5° (440) two theta degrees, supporting the presence of the spinel structure of CoFe_2_O_4_, NiFe_2_O_4_, MgFe_2_O_4_ and MnFe_2_O_4_ (PDF 22-1086; PDF: 54-0964; PDF 36-0398; PDF 74-403) (Figure 2B). Other oxide phases were not identified on the XRD patterns of the ferrites, suggesting that the synthesis of pure spinel nanoparticles was successful in all cases.

The size and morphology of the magnetic nanoparticles were characterized by high-resolution transmission electron microscopy (HRTEM), suggesting a spherical morphology of the synthetized nanoparticles (Figure 3A–D). The secondary structure of the magnetic nanospheres is visualized in Appendix A. The spherical aggregates consisted of 4–10 nm particles organized into spherical particles due to their surface functional groups. This self-assembly was also observed for all four amine-functionalized ferrites. 

The detailed size statistics from the TEM results are more informative than the mean particle size, which was calculated based on Sherer’s method, as the latter results in the calculation of the average size of the individual particles which build up the nanospheres (Table 1). On the FTIR spectra of the ferrites, carbon-containing functional groups were identified, originating from the adsorbed ethanolamine and ethylene glycol. Thus, CHNS element analysis was performed to determine the exact quantification of the carbon and nitrogen (Table 1). The carbon content of the magnetic particles was found between 1.7 and 6.4 wt%. The lowest carbon content was found in the sample of the manganese ferrite (1.7 wt%), while the highest nitrogen content was measured in the nickel ferrite sample, indicating the presence of amine functional groups.

### 2.2. The Application of NH_2_-Functionalized Magnetic Nanoparticles for Glyco-Analytical Sample Preparation

After the synthesis and characterization of the magnetic nanoparticles, their suitability was tested for the selective enrichment of fluorescently labeled glycans. PNGase F-digested serum samples were labeled by procainamide via reductive amination in triplicate and purified by CoFe_2_O_4_-NH_2_, NiFe_2_O_4_-NH_2_, MgFe_2_O_4_-NH_2_ and MnFe_2_O_4_-NH_2_, as described in our previous study [24]. Briefly, 200 µL of 20 mg/mL magnetic nanoparticles in water was transferred into a 1.5 mL tube and placed onto a magnetic holder to remove the supernatant. Then, the labeled glycan sample was mixed with the nanoparticles in 90% acetonitrile and 10% water. The purified sample was eluted from the nanoparticles using 100 µL of 100% water and analyzed by HILIC-UPLC. In our results, we found that each synthetized nanoparticle was suitable for glycan purification. Integrating the chromatograms, we found that NiFe_2_O_4_-NH_2_ and MgFe_2_O_4_-NH_2_ provided the highest signal intensities (Appendix A), while the lowest free dye peak was obtained from the NiFe_2_O_4_-NH_2_-purified sample (Appendix A). Based on these results, we decided to use NiFe_2_O_4_-NH_2_ for further experiments. Next, we evaluated the impact of the nanoparticle concentration on the observed fluorescence intensity using 0.5 mg/mL, 1 mg/mL, 2 mg/mL, 4 mg/mL, 8 mg/mL and 16 mg/mL concentrations of NiFe_2_O_4_-NH_2_ for the purification of glycans released from 9 µL serum. In our results, we found that from 0.5 mg/mL to 4 mg/mL concentrations, the intensities linearly increased, while from 4 mg/mL to 16 mg/mL, they linearly decreased (data not shown). This can be explained by the fact that the 4 mg/mL concentration is sufficient to capture all the glycans in the sample, although exceeding this critical concentration can result in the loss of sugars due to the self-assembly of spherical aggregates and problematic pipetting. Next, we evaluated the reproducibility of the glycan sample purification by 4 mg/mL NiFe_2_O_4_-NH_2_ by preparing six parallel serum samples. As shown in Appendix A, analyzing the area percentage of the 43 peaks of the six individual replicates, the variation coefficient was 3.52 on average, suggesting excellent relative peak area percentage reproducibility, which is crucial in clinical studies. Based on these findings, we applied the developed approach to analyze the serum *N*-glycome of 38 patients diagnosed with multiple sclerosis and corresponding healthy controls by hydrophilic interaction liquid chromatography with fluorescence and mass spectrometric detection, as visualized in Appendix A. During the LC analysis, 43 individual glycan peaks were quantified in triplicate, generating 228 chromatograms. In the statistical analysis, the abundance of 15 structures was found to be significantly altered in MS patients, as listed in Appendix A. The identification of significant differences was obtained from mass spectrometric data, which are included in Appendix A. The main significant alterations were identified as FA2 (retention time: 16.07, *m*/*z*: 842.11^2+^), FA2G2S1 (retention time: 25.89, *m*/*z*: 1149.74^2+^), A2G2S1 (retention time: 24.57, *m*/*z*: 1076.77^2+^) and FA2BG2S1 (retention time: 26.90, *m*/*z*: 1251.62^2+^), as visualized in Figure 4A–D. It is important to note that the significant alterations were only observed between the female groups.

Similarly, the linear discriminant analysis showed a slight overlap between the male groups, while the healthy females were clearly separated from MS females based on their serum *N*-glycome distribution (Figure 5). The structures contributing to the separation of the patient groups were also similar to the significantly altered glycans presented in Figure 4, as FA2, A2G2S1 and FA2BG2S1 were clearly prominent. These findings are in agreement with previous studies, as altered glycosylation of plasma proteins was identified by Cvetko et al. [13]. In their study, they found significantly increased sialylation levels on the structures of A3G3S2, A3S3S3, A4G4S3, A4G4S4 and FA3G3S3 in MS patients. This is partly in agreement with our results, which also found increases in A3G3S2 and A4G4S3 (Appendix A). Moreover, A2G2S1 and FA2BG2S1 were also higher, but only in female MS patients. They also found lower FA2 levels from IgG-derived traits, which is identical to our findings, which showed that decreased FA2 was one of the main contributing factors in the separation of female MS patients. Similar to our results, a high mannose level on the Man5 structure was reported, although in our dataset, the Man6 structure was also significantly higher (Appendix A). Kennedy and colleagues also reported higher sialylation of serum-derived proteins in MS compared to healthy controls [14]. The level of galactosylation was also increased in MS, similar to our results, as A2G2 and FA2BG2 glycan ratios were higher in the disease groups, although this was only significant between the female patients.

## 3. Materials and Methods

### 3.1. Chemicals

Formic acid, ammonium-hydroxide, acetic acid, acetonitrile, picoline borane, procainamide-hydrochloride, dimethyl sulfoxide, magnesium(II) nitrate hexahydrate and ethanolamine were purchased from Sigma-Aldrich (St. Louis, MO, USA). PNGase F was obtained from New England Biolabs (Ipswich, MA, USA). Nickel(II) nitrate hexahydrate and sodium acetate were obtained from Thermo Fisher Scientific (Kandel, Germany). Mangan(II) nitrate tetrahydrate was provided by Carl Roth (Karlsruhe, Germany). Cobalt(II) nitrate hexahydrate and iron(III) nitrate nonahydrate were purchased from VWR International (Leuven, Belgium).

### 3.2. Preparation of the Amine-Functionalized Ferrite Nanoparticles

Amine-functionalized ferrite samples were synthesized by a modified coprecipitation method. In 50 mL ethylene glycol, 8.08 g (2 mmol) iron(III) nitrate nonahydrate and nitrate salts of the other appropriate metals (2.91g Co(NO_3_)_2_∙6H_2_O, 2.91 g, Ni(NO_3_)_2_∙6H_2_O, 2.51 g Mn(NO_3_)_2_∙4H_2_O and 2.56g Mg(NO_3_)_2_∙6H_2_O) were dissolved. Sodium acetate (12.30 g, 15 mmol) was dissolved in 100 mL ethylene glycol and heated to 100 °C in a three-neck flask under reflux with continuous stirring. The solutions of metal ions were added to the glycol-based sodium acetate solution, after the addition of 35 mL ethanolamine. After 12 h of continuous agitation and reflux, the cooled solution was separated by centrifugation (4200 rpm, at 10 min). The solid phase was washed with distilled water several times, until the magnetic ferrite was easily separated by a magnet from the aqueous phase. Finally, the ferrite was rinsed with absolute ethanol and dried by lyophilization. The ferrites formed according to Equation (1).
(1)$$M^{2+}+2{\;\mathrm{Fe}}^{3+}+8{\;\mathrm{OH}}^{-}\overset{∆}{\to }{\;\mathrm{MFe}}_{2}O_{4}+4{\;H}_{2}O\;\left(M:\;\mathrm{Ni},\;\mathrm{Mg},\;\mathrm{Co},\;\mathrm{Mn}\right)$$

### 3.3. Characterization of the Amine-Functionalized Ferrite Nanoparticles

The examinations of the particle size, morphology and electron diffraction pattern of the ferrite samples were carried out by high-resolution transmission electron microscopy (HRTEM) using a Talos F200X G2 electron microscope (FEI, Hillsboro, OR, USA) with a field-emission electron gun, X-FEG (accelerating voltage 20–200 kV). For the imaging and electron diffraction, a SmartCam digital search camera (Ceta 16 Mpixel, 4 k × 4 k CMOS camera) was used with a high-angle annular dark-field (HAADF) detector. For identification of the ferrite phases, X-ray diffraction (XRD) measurements were applied using a Bruker Discovery diffractometer (Cu-Kα source, 40 kV and 40 mA) in parallel beam geometry (Göbel mirror) with a Vantec detector, using Powder Diffraction Files (PDFs). The average crystallite size of the oxide domains was calculated by the mean column length calibrated method, using the full width at half maximum (FWHM) and the width of the Lorentzian component of the fitted profiles. For the evaluation, TOPAS 4 software was used. The surface functional groups of the ferrite nanoparticles were identified with Fourier-transform infrared spectroscopy (FTIR) by a Bruker Vertex 70 spectrometer in transmission mode. During the FTIR study, a 10 mg ferrite sample was pelletized with 250 mg spectroscopic-grade potassium-bromide. The carbon content of the ferrite samples was measured by a Vario Macro CHNS elemental analysis instrument using phenanthrene as a standard (C: 93.538%, H: 5.629%, N:0.179%, S: 0.453%) from Carlo Erba Inc. The carrier gas was helium (99.9990%), while oxygen (99.995%) was used for oxidation, and the samples were loaded into tin foils.

### 3.4. Patient Samples

Serum samples from 38 control patients (average age 48.8) and 38 patients with multiple sclerosis (average 48.6) were collected in the Department of Neurosurgery at the Borsod-Abaúj-Zemplén County Center Hospital and University Teaching Hospital (Miskolc, Hungary). Informed consent forms were signed by all the patients in accordance with the Declaration of Helsinki. The study was approved by the regional research ethics committee (Ethical approval number: RKEB/IKEB-G-102-102-2018).

### 3.5. N-Glycan Release, Labeling and Clean-Up

The glycan release was performed using 9 µL of serum sample, according to the PNGase F deglycosylation protocol of New England Biolabs (Ipswich, MA, USA). The released glycans were labeled by the addition of 10 μL 0.3 M procainamide and 300 mM picoline borane in 70%/30% dimethyl sulfoxide/acetic acid and incubated for 4 h at 65 °C. After the PNGase F digestion and fluorescent labeling, the sample was suspended with the magnetic nanoparticles in 90% acetonitrile. As these particles are hydrophilic, they create hydrogen bonds with water molecules; however, when the water concentration is decreased (90% acetonitrile, 10% water), they connect to other hydrophilic molecules such as sugars. At the end of the purification, the carbohydrates can be eluted from the magnetic nanoparticles by water and analyzed by LC.

### 3.6. UPLC-FLR-MS Analysis

The prepared *N*-glycans were analyzed using a Waters Acquity ultra-performance liquid chromatography system equipped with a fluorescence detector and a single quad mass detector. The system was controlled by the Empower 3 chromatography software (Waters, Milford, MA, USA). Separations were performed with a Waters BEH Glycan column, 100 × 2.1 mm i.d., 1.7 μm particles, using a linear gradient of 72–55% acetonitrile (Buffer B) at 0.4 mL/min in 42 min, with 50 mM ammonium formate pH 4.4 as Buffer A. Samples were dissolved in 75%/25% acetonitrile/water and 5 μL was injected in all runs. The sample manager temperature was set at 15 °C, while the column temperature was 60 °C during the runs. The fluorescence detection excitation and emission wavelengths were λex = 309 nm and λem = 359 nm. During the MS analysis, a 2.2 kV electrospray voltage was applied to the capillary. The desolvation temperature was set to 120 °C, while the desolvation gas flow was 500 L/h. Mass spectra were acquired over the range of 500–2000 *m*/*z* in positive ionization mode and data-independent acquisition.

### 3.7. Data Analysis

All patient samples were analyzed in triplicate, and all chromatograms were quantified by Empower 3 chromatography software (Waters, Milford, MA, USA). The data analysis was performed in IBM SPSS Statistics 25, and Past 4.02. GlycoWorkBench was used for the mass calculation of the individual glycan structures. Glycan nomenclature was used as described by Harvey et al. [25].

## 4. Conclusions

In this study, NH_2_-functionalized magnetic nanoparticles were synthetized, characterized and applied for the *N*-glycomic analysis of serum samples. The suitability of CoFe_2_O_4_-NH_2_, NiFe_2_O_4_-NH_2_, MgFe_2_O_4_-NH_2_ and MnFe_2_O_4_-NH_2_ was compared for glycan purification, suggesting that NiFe_2_O_4_-NH_2_ is the most efficient. The optimal nanoparticle concentration was also evaluated and resulted in excellent reproducibility. Using this in-house synthetized magnetic nanoparticle-based approach, we analyzed the serum *N*-glycome of patients with MS compared to healthy individuals. Higher sialylation, galactosylation and mannose levels were found in MS patients, although these alterations were only significant between the female groups. Our plan going forward is to automate the sample preparation process using a liquid handling robot in combination with the developed NiFe_2_O_4_-NH_2_-based approach, allowing high-throughput and large-scale glycomics studies.

## Figures and Tables

**Figure 1 ijms-23-09095-f001:**
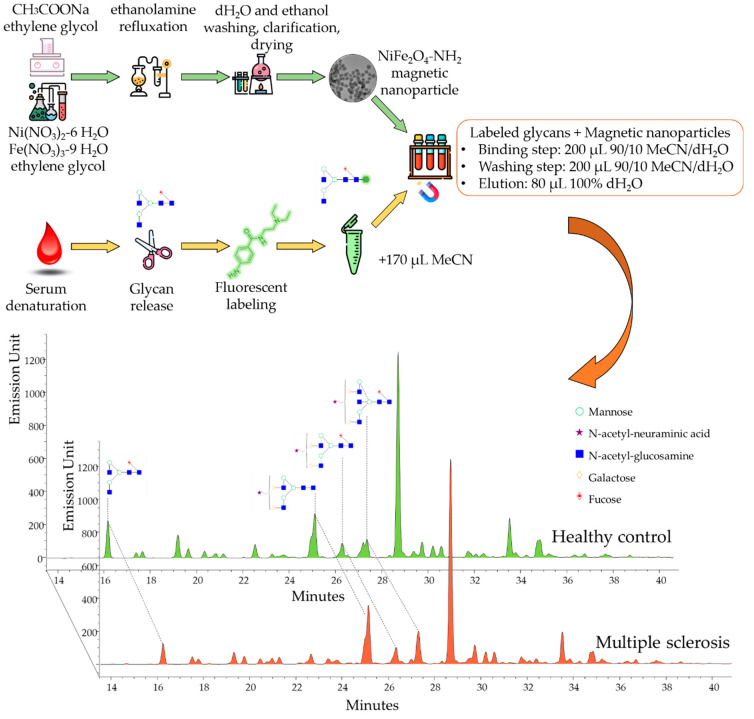
Scheme of the magnetic nanoparticle-based sample preparation for the *N*-glycomic analysis of patients with MS.

**Figure 2 ijms-23-09095-f002:**
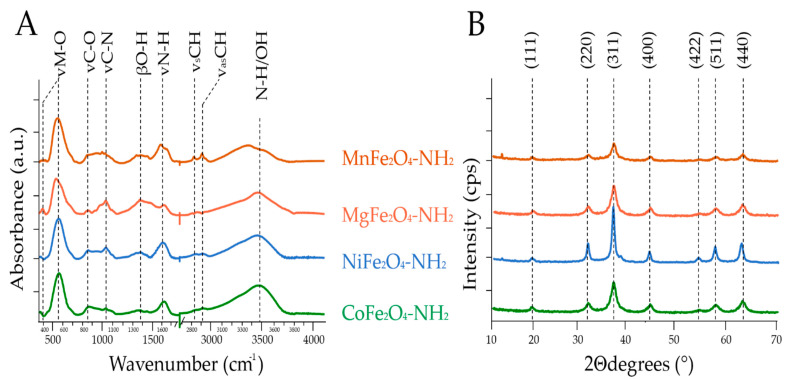
FTIR (**A**) and XRD (**B**) analysis of the synthetized nanoparticles.

**Figure 3 ijms-23-09095-f003:**
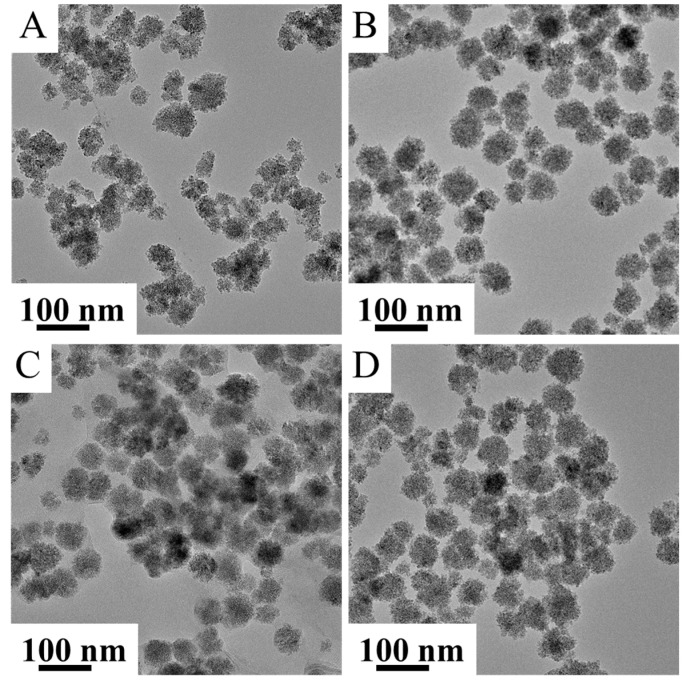
HRTEM picture of the CoFe_2_O_4_-NH_2_ (**A**), NiFe2O_4_-NH_2_ (**B**), MgFe_2_O_4_-NH_2_ (**C**) and MnFe_2_O_4_-NH_2_ (**D**) magnetic nanoparticles.

**Figure 4 ijms-23-09095-f004:**
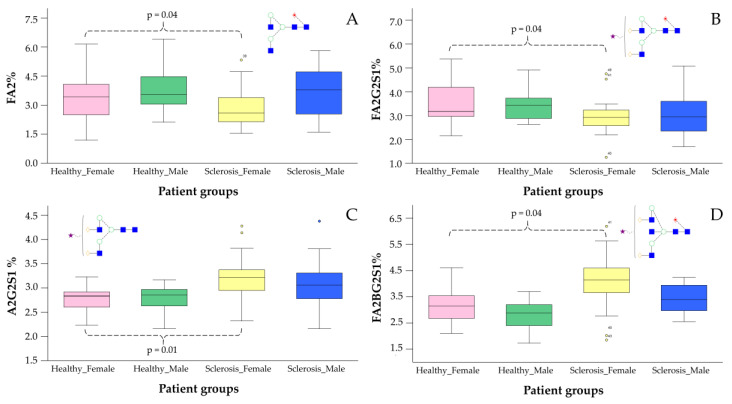
Significant glycosylation alterations between healthy controls and female MS patients.

**Figure 5 ijms-23-09095-f005:**
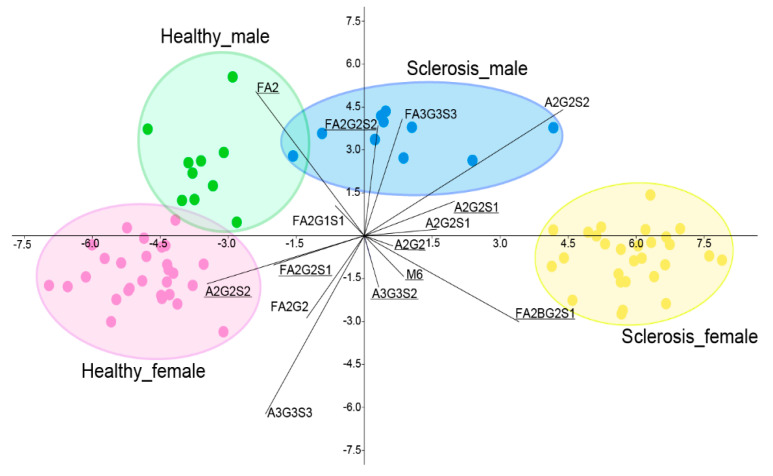
Linear discriminant analysis of female/male MS patients and healthy controls based on their serum *N*-glycome pattern (significant alterations are underlined).

**Table 1 ijms-23-09095-t001:** Carbon content, specific surface area and average particle size (based on XRD) of the ferrite nanoparticles.

	NiFe_2_O_4_-NH_2_	MnFe_2_O_4_-NH_2_	MgFe_2_O_4_-NH_2_	CoFe_2_O_4_-NH_2_
C content (wt%)	6.3	1.7	6.4	2.5
N content (wt%)	1.4	0.3	0.6	0.4
Surface area (m^2^/g)	93.8	155	86.3	279.4
Particle size (nm)	6 ± 2	8 ± 1	6 ± 1	4 ± 2

## Data Availability

The generated data can be requested from the corresponding author.

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
