# Peer review of "NH2-Functionalized Magnetic Nanoparticles for the N-Glycomic Analysis of Patients with Multiple Sclerosis"

_ijms, 2022, doi:10.3390/ijms23169095_

Round 1

Reviewer 1 Report

Manuscript titled “NH2-Functionalized Magnetic Nanoparticles for The N-glycomic Analysis of Patients with Multiple Sclerosis” by Dalma Dojcsak and co-workers describe a method for synthesizing, characterizing and analyzing amine functionalized nanoparticles that was used for the glyco-analysis of serum samples of patients with MS. Specifically the authors performed and looked at the N-glycome of these samples. MS affects about 40,000 people in the US alone (https://www.ncbi.nlm.nih.gov/pmc/articles/PMC6396336/). Though its etiology is debated, it is known that genetic and environmental factors trigger the development of MS. Though MRI imaging is used for diagnosis however this does not allow for individual characterization and prediction of MS. Given that currently there are no molecular biomarkers, this study attempts to provide a mechanistic understanding by examining the N-glycome of the patient serum samples and mitigate the lacking biomarker gap. Having said that, I found the parts of the manuscript a bit disjointed & difficult to navigate and understand/follow. For example under results & discussion: on page 6, after talking about their result of finding 43 peaks of which 15 were altered in MS samples, the authors jump in to talking about a figure which cannot be found and the results obtained from that analysis. It would definitely help to have the right figure & moreover it would help to have some context before diving into the results – like which are the 15 glycans that are altered. And then start by saying that linear discriminant analysis was done which revealed the following. Also, I suggest that authors discuss the statistics method a bit more in detail. I have some more changes suggested and questions, once these are addressed the manuscript may be considered for publication.

Specific Comments: 

1)     On page 3, line number 90-91, authors have mentioned that “In the infrared spectrums, two bands were identified originating from the tetrahedral 90 (between 500 cm-1 and 600 cm−1 wavenumbers),”, are these 2 bands or is it a single band with a shoulder, or unresolved feature?

2)     Also I suggest that authors use minor tick marks to denote the wavenumbers between 500 wavenumbers – this will make it easier for readers to follow the text. For example, authors have discuss a band at 600 cm-1, which is difficult to make out.

3)     Though FTIR is not my area of work, I am curious to know whether the bands at 800 and 900 cm-1, as claimed to be arising from C-O stretch of alcoholic groups of the ethylene glycol- they look pretty broad, is that an effect of H-bonding ? Do authors have a feel for why these bands are so broad?

4)     Have the authors noticed any shifting (red/ blue) of bands, especially the N-H / O-H region ?

5)     How confident are the authors in their assignment of bands especially the low intensity ones? 

6)  Rephrase sentence 154 on page 5, because it sounds like the glycans in serum samples were purified by addition of nanoparticles – which is not the case.

2)     Similarly, in lines 158-159, what do the authors mean by “purified sample”- please clarify that this is the glycan serum sample with nanoparticle which was purified.

3)      Supporting figure 3A, B are confusing- define what the titles are on Y-axes in the figure. To me, it seems that Figure 3A has the glycan, because the labeled glycans are shown next to it. So which is the figure with procainamide only?

4)     For the concentration dependence on fluorescence experiments, lines 165-172, authors should either add a figure in supporting info or say that they have not shown the data here in the main text, because if not mentioned, readers may be looking for it.

5)     In the last row of supporting info Table 1, is the mean calculated of the CVs (entry in column4) and SDs  (entry in column 3)?

6)     What are the 43 peaks of?

7)     In lines 186-189 on page# 6, authors have referred to Figure 3A-D, which is incorrect because according to manuscript, figure 3 are HRTEM figures. I think that authors may be referring to figure4, either ways please address this discrepancy & refer to the correct figure.

8)     In conclusions, the authors claim that optimal nanoparticle concentration was also evaluated resulting excellent reproducibility (line 222), I am not sure about the reproducibility part, please clarify how this was obtained.

9)     Authors should also rephrase the line 226: though authors do mention that the alterations discovered were only in females, and not male patients, they should rephrase it this line to put it in active form and may be remove the 1st part of the line because the 1st part is misleading because alterations were found only for female. Also do the authors have a handle on why alterations were not significant in male samples?

10)   Under the mass spec method section, authors should describe details of the mass spect method used- was it a DDA or DIA?

Reviewer 2 Report

The article “NH2-Functionalized Magnetic Nanoparticles for The N-glycomic Analysis of Patients with Multiple Sclerosis” by Dojcsák and colleagues investigates the use of different types of magnetic nanoparticles for purifying hydrolysed glycans samples from healthy vs multiple sclerosis patients.

The article might be of interest to the researcher if the method is properly set-up and tested.

INTRODUCTION

The authors should stress the importance of the glycomic analysis as a potential diagnostic tool, e.g. including https://www.mdpi.com/2218-1989/11/9/566 and https://www.mdpi.com/2073-4409/11/9/1575.

CHARACTERIZATION

The average size of nanoparticles has to be also determined by DLS (dynamic light scattering) for quantifying the hydrodynamic radius in solution.

In addition to the FT-IR analysis, a quantitative determination of the -NH2 groups on the surface of the nanoparticles must be performed, e.g. by a simple ninhydrin test. This will also allow to gain better understanding of the glycan purification step, e.g. when comparing the values obtained by the different nanoparticles.

MULTIVARIATE STATISTICAL ANALYSIS

The authors could use the obtained data for preparing and testing a predictive model based on the ROC analysis.

OTHER POINTS

Some English and editing errors are present and must be corrected.

N- in chemical names goes in italics (line 2, 58, 71, 85, 178, 198, 290, etc).

ml must be corrected to mL (the same applies to ul, etc), e.g. in Figure 1.

Round 2

Reviewer 1 Report

11)   On page 3, line number 90-91, authors have mentioned that “In the infrared spectrums, two bands were identified originating from the tetrahedral 90 (between 500 cm-1 and 600 cm−1 wavenumbers),”, are these 2 bands or is it a single band with a shoulder, or unresolved feature?

The band between 500 cm-1 and 600 cm-1 is one convoluted band with a shoulder. Two bands belong to the mentioned metal-oxigen bonds, the one at 433 cm-1 and another band at 576 cm-1 is visible, these vibrations were signed as νM-O on the FTIR spectrums.

I suggest that the authors add the above info/ explanation to the revised version.

22) Though FTIR is not my area of work, I am curious to know whether the bands at 800 and 900 cm-1, as claimed to be arising from C-O stretch of alcoholic groups of the ethylene glycol- they look pretty broad, is that an effect of H-bonding ? Do authors have a feel for why these bands are so broad?

The mentioned broad band is so wide due to the convolution of several vibration bands. At the start of the fingerprint region (400-1600 cm-1) are located several vibration band (i.e.: νC-N, νC-O, ω(wagging)N-H, δCH2). It would be possible to deconvolute the broad band, but it would be uncertain to identify all individual bands.

Again I suggest that authors add this to the manuscript, that way readers have no doubts.

3) Rephrase sentence 154 on page 5, because it sounds like the glycans in serum samples were purified by addition of nanoparticles – which is not the case.

Actually that is perfectly the case. After the PNGase F digestion and fluorescent labeling the sample was suspended with the magnetic nanoparticles in 90% acetonitrile. As these particles are hydrophilic, they like to create hydrogen bonds with water molecules although when we decrease the concentration of the water (90% acetonitrile 10% water) they connect to other hydrophilic molecules such as sugars. At the end of the purification we can just simply elute the carbohydrates from the magnetic nanoparticles by water.

It would help if the authors added this explanation at the method section to remove/take care of any doubts like mine

Reviewer 2 Report

Although the authors did not address all the points raised, the revised version of the article is improved compared to its previous one and, in my opinion, deserves publication in IJMS.

Author Response

Thank you for your suggestion.